# Mathematical Models for Machining Optimization of Ampcoloy 35 with Different Thicknesses Using WEDM to Improve the Surface Properties of Mold Parts

**DOI:** 10.3390/ma16010100

**Published:** 2022-12-22

**Authors:** Katerina Mouralova, Josef Bednar, Libor Benes, Tomas Prokes, Radim Zahradnicek, Jiri Fries

**Affiliations:** 1Faculty of Mechanical Engineering, Brno University of Technology, 616 69 Brno, Czech Republic; 2Faculty of Production Technologies and Management, Jan Evangelista Purkyně University, 400 96 Ústí nad Labem, Czech Republic; 3Department of Machine and Industrial Design, VSB - Technical University of Ostrava, 708 00 Ostrava, Czech Republic

**Keywords:** WEDM, surface topography, cutting speed, Ampcoloy, design of experiment, machining parameters

## Abstract

Wire electrical discharge machining (WEDM) is an unconventional machining technology that can be used to machine materials with minimum electrical conductivity. The technology is often employed in the automotive industry, as it makes it possible to produce mold parts of complex shapes. Copper alloys are commonly used as electrodes for their high thermal conductivity. The subject of this study was creating mathematical models for the machining optimization of Ampcoloy 35 with different thicknesses (ranging from 5 to 160 mm with a step of 5 mm) using WEDM to improve the surface properties of the mold parts. The Box–Behnken type experiment was used with a total of 448 samples produced. The following machining parameters were altered over the course of the experiment: the pulse on and off time, discharge current, and material thickness. The cutting speed was measured, and the topography of the machined surfaces in the center and at the margins of the samples was analyzed. The morphology and subsurface layer were also studied. What makes this study unique is the large number of the tested thicknesses, ranging from 5 to 160 mm with a step of 5 mm. The contribution of this study to the automotive industry and plastic injection mold production is, therefore, significant. The regression models for the cutting speed and surface topography allow for efficient defect-free machining of Ampcoloy 35 of 5–160 mm thicknesses, both on the surface and in the subsurface layer.

## 1. Introduction

Wire electrical discharge machining (WEDM) is an unconventional machining technology that is vital for many industries, such as the food, aerospace, and automotive industries. WEDM is suitable for the machining of all materials with at least minimum conductivity. That offers a lot of benefits, especially the possibility of machining materials which are conventionally nonmachinable. WEDM makes it possible to machine highly tough, soft, or very hard materials, where otherwise the need to purchase a conventional tool would prove very expensive. An increased energy consumption counts amongst the few WEDM disadvantages; however, optimizing the process using one of the optimization techniques [1,2] can help to reduce the aforementioned. Unfortunately, it is not possible to set the machining speed as WEDM does not work like other conventional machining technologies. The machine controls set the speed automatically based on the machining parameters. The tool (wire) must never come into direct contact with the workpiece. For this purpose, the machine manufacturer supplies technologies (the summary of the machining parameters) for different material types and thicknesses (the usual step is 5 mm). Thickness plays a very important role here as well and must always be taken into account [3,4]. The main disadvantages of WEDM technology are the low speed of material removal and remelted layer formation.

The advantages of Ampco alloys include a high thermal conductivity of up to 208 W·m^−1^·K^−1^ and relatively good hardness of up to 450 HB [5]. These properties make them ideal for the production of parts for plastic injection molds or the mold itself [6]. WEDM technology is vital for the manufacturing of mold parts and molds themselves as traditional machining is sometimes entirely impossible.

Wu et al. [7] studied how the geometric structure, such as sinusoidal and rectangular shapes, of certain metals influences their wettability. Cu alloys used in the research developed micro- and nano-scale craters during wire electrical discharge machining (WEDM). The spark discharge caused rapid remelting, resulting in the formation of a recast layer on the alloy. Its surface exhibited an improved hydrophobicity due to the formation of multiscale submillimeter structures. The sinusoidal structures showed better hydrophobicity than the rectangular ones. The servo voltage during WEDM can significantly affect the shape and size of the micron-scale structures on the surface. Ahmed et al. [8] found that to maximize the surface area for optimal heat transfer, there has to be the largest number possible of microchannels. The interchannel fin thickness (IFT) represents the most important feature influencing this number. However, it is difficult to fabricate deep microchannels through conventional methods, which is why WEDM could be the solution. In the study, the minimum IFT and the machining parameters were investigated for copper microchannels. Satishkumar et al. [9] studied the metal removal rate (MRR) and surface roughness of oxygen-free high thermal conductivity (OFHC) copper machined by WEDM. The Taguchi method was used to develop mathematical models to improve the said metal removal rate and surface roughness. They concluded that the pulse on time, pulse off time, and wire tension are the most important factors influencing MMR and surface roughness. The found solutions can serve as the recommended machining parameters. Li [10] investigated the optimal machining parameters for copper electrode WEDM manufacturing. The parameters’ influence on cutting speed and surface roughness was studied to achieve the best surface quality. Aspects such as the discharge energy, formation of adhesive on the cutting surface, electric erosion products, and their influence were analyzed using analytical methods. Tests proved the pulse width and peak current to be the most influential factors for surface roughness and cutting speed, respectively. Sathiyaraj et al. [11] studied different WEDM machining parameters and how different combinations of the pulse on time, pulse off time, and peak current influence the MRR and surface roughness (Ra). Taguchi’s L9 orthogonal array, advantageous for its need for a small number of trials, was successfully employed in the experiment and, along with the analysis of variance (ANOVA), helped determine the importance of the aforementioned machining parameters. Meenakshi et al. [12] studied the surface roughness of copper metal matrix composite (MMCs) machined with WEDM. The experiment was conducted using Taguchi’s L9 orthogonal array. Employing ANOVA and a multilinear regression model (MLRM), they established the parameters’ significance for surface roughness. The MLRM coefficients were adjusted using an artificial immune system algorithm. Optimizing the machining parameters such as the spark on time, spark off time, peak current, and wire feed, the surface roughness was reduced. Evran [13] investigated the effects of machining parameters on surface roughness (SR) and MMR in the WEDM of hard copper alloy. The parameters in question were the wire type, pulse time, and duration between two pulses. The Taguchi method was used for the experiment’s design. The signal-to-noise ratio and variance analysis were employed to determine the parameters’ effect and their per cent contributions regarding SR and MRR, respectively. Wang et al. [14] simulated how the discharge craters formed on a beryllium copper alloy surface after electrical discharge machining (EDM) affect its drag reduction. The topological features of the surface under different EDM parameters were extracted using power spectrum and wavelet analysis, and the resulting mathematical models were used to simulate the interaction of water with the surface using Fluent. This showed the drag reduction performance for different machining currents, pulse width, and discharge energy settings. The results were verified using a real test experiment.

Mahapatra et al. [15] studied WEDM machining factors and their influence on performance characteristics. The parameters—discharge current, pulse duration, pulse frequency, wire speed, wire tension, and dielectric flow—were analyzed using Taguchi’s parameter design. The influence of the said parameters on the resulting responses—MRR, surface finish (SF), and cutting width (kerf)—was observed. A mathematical model was proposed to adjust the parameters in order to optimize all responses.

Chiang et al. [16] studied how to optimize the WEDM process of Al2O3 particle-reinforced material (6061 alloy). Using grey relation analysis, the influence of the machining parameters on the performance characteristics was established. The studied parameters were the cutting radius, on and off time, arc on and off time, servo voltage, wire feed, and water flow. The performance characteristics considered were the material removal rate and surface roughness. The study proposed a relevant approach for the characteristics’ improvement.

WEDM optimization is a complex task as there is a large number of machining parameters, and each material’s thickness (usually 5 mm) needs to be optimized separately. This optimization will, however, bring significant savings in the machining time and hence reduce the amount of electrical energy consumed to machine a particular part. In order to gain a detailed understanding of WEDM, extensive studies have been conducted on the machining of materials such as the abrasion-resistant steels Hardox 400 [17] and Creusabro 4800 [18], aluminium alloy 7475-T7351 [19], and pure molybdenum [20]. The purpose of this study was to optimize the WEDM process for the machining of Ampcoloy 35 in terms of maximizing the cutting speed and surface quality for material thicknesses of 5–160 mm and a comprehensive study of the surface and subsurface area of all machined samples. What makes this study unique is the large number of the tested thicknesses, ranging from 5 to 160 mm with a step of 5 mm. A total of 32 different thicknesses were tested, with 448 samples produced. The contribution of this study to the automotive industry and plastic injection mold production is, therefore, significant.

## 2. Experimental Setup and Material

### 2.1. Experimental Material

The samples for the experiment were made of copper alloy Ampcoloy 35. The chemical composition of the material is defined by the standard in wt.%—4% Zn, 7% Sn, 6% Pb, and Cu balance. Ampcoloy 35 is an alloy with very good thermal conductivity, which is 45 W·m^−1^·K^−1^. Due to its higher tin content, the tensile strength reaches 250 MPa. Ampcoloy alloys are mainly used for the production of replaceable parts and mold parts that need to be changed due to wear. The initial cube-shaped semiproducts were used for the experiment, an example of which is shown in Figure 1a. Each sample was cut to a length of 3 mm, as shown on the sample in Figure 1b. There were always exactly 6 samples on each plate, and the first sample was marked with a chamfered edge for a better orientation on the plates. A large number of plates with different thicknesses from 5 to 160 mm with a 5 mm step (Figure 1c) was produced, resulting in a total of 32 different material thicknesses examined. The samples’ microstructure and chemical composition analysis (EDX) are shown in Figure 1d.

### 2.2. WEDM Machine Setup

A Robocut C400iB EDM wire-cutting machine from FANUC, shown in Figure 2a, was used to produce samples. This machine is equipped with CNC control in all 5 axes, which allows for the production of conical shapes. The machine can produce workpieces of 730 × 630 × 250 mm in size and a maximum weight of 500 kg. The travel of the X- and *Y*-axis is 400 × 300 mm and 255 mm of the *Z*-axis. Machining is possible at a maximum of ±30° angle. The workpiece was immersed in a dielectric bath of nonionized water at all times. During the entire machining time, flushing was ensured by the upper and the lower jets, with both of them placed as close to the workpiece as possible. The tool electrode was a 0.25 mm brass wire supplied by PENTA under the designation PENTA CUT P. The strength of the wire was 1000 N/mm^2^. The sample production is shown in Figure 2b,c.

The key output characteristics describing EDM wire cutting are the cutting speed and quality of the machined surface. These will be assessed both in the center of the sample and at its margin. The removal of eroded particles is expected to be poorer in the samples’ center, as the dielectric access to the cutting point is more difficult. As a result, the surface quality will deteriorate. The dependence of these characteristics on the machining parameters was modelled using a design of experiment (DoE). The ground for this large-scale design of experiment is one replication of the Box–Behnken model with two replications of the center point (Table 1) for the input factors discharge current (*I*), pulse on time (*T_on_*), and pulse off time (*T_off_*), and in the ranges according to Table 2.

These parameters and ranges were selected based on extensive previous tests. The screening test proved the parameters of gap voltage and wire feed rate to be negligible, which is why they were set to a constant value, that is, 60 V for the gap voltage and 12 m/min for the wire feed rate. The thicknesses (*t*) of the workpieces varied from 5 to 160 mm in steps of 5 mm, i.e., 5, 10, 15… mm, and 14 samples of each thickness were produced, for a total of 448 samples of different thicknesses. There were 32 thicknesses in total, so the experiment entailed 32 repeated measurements that determined the reproducibility of the measurements of the investigated responses in the consecutive regression models. The wire did not break at any thickness or machining parameters.

### 2.3. Experimental Methods

Samples produced in the design of the experiment on an EDM wire-cutting machine were cleaned in an ultrasonic cleaner from Sonic (Morinville, Alberta Canada) and subjected to a comprehensive analysis using an LYRA3 scanning electron microscope (SEM) from Tescan (Brno, Czech Republic). This microscope also included an energy-dispersive X-ray detector, which enabled the analysis of the chemical composition. In order to study the surface and subsurface layers, metallographic preparations were made, which enabled the representation of cross-sections of the sample. The specimens were prepared using conventional techniques, wet grinding and diamond paste polishing using the automatic TEGRAMIN 30 preparation system from Struers (Westlake, Cleveland, OH, USA). The final mechanical–chemical polishing was carried out using Struers OP-Chem suspension. Once etched with aqua regia 1:20 (HCL:HNO_3_), the structure of the material was observed using an Axio Observer Z1m inverted light microscope (LM) from ZEISS and documented. The topography and 3D relief of the machined samples’ surface were studied using a noncontact 3D profilometer IFM G4 supplied by Alicona based on the principle of coherence correlation interferometry. Finally, 3D surface reliefs were created using a contact 3D profilometer Dektak XT from Bruker and further analyzed using Gwyddion software.

## 3. Results and Discussion

### 3.1. Statistical Evaluation of Surface Topography and Cutting Speed

The evaluated surface topography parameter, in this case, was the arithmetical mean deviation of the profile (Ra), which is the main topography parameter that is usually considered. Due to the large number of analyzed samples, only one parameter was evaluated. The measurement was performed according to the standard for profile parameters ISO 4287 [21]. The first spot chosen for the measurement was always 0.1 mm from the margin of the samples, and the second was its center. These two spots were chosen based on the knowledge that, in the center, the dielectric fluid is not distributed as well as at the margins, and the drainage of the eroded material is not as smooth as at the margins, which always makes the topography parameters higher. The measurement at the margin is further designated as the “margin” and the measurement at the center as the “center”. We can see higher values registered in the centers of the samples than at their margins. The maximum Ra value of 4 µm was measured in the centers and only 3.6 µm at the margins. These values are higher than the maximum Ra values observed in a previous study performed by Mouralova et al. [22] when machining the same material of a 10 mm thickness. This result was to be expected, as with increasing workpiece thickness, the effect of poor dielectric supply to the sample’s center has to show. On the contrary, the lowest value of Ra 1.6 µm was observed in Sample 52, machined with the following parameters: *T_on_* = 6 µs, *T_off_* = 40 µs, and *I* = 25 A. Similarly low values were obtained for copper machining in the studies undertaken by Venkateswarlu et al. [23] and Satishkumar et al. [9]. The thickness of the machined material in these studies was, however, only 10 mm.

For each sample, the cutting speed was read from the machine’s display. True to our expectations, the highest speed of 20.24 mm/min was observed in sample No. 7 with the lowest material thickness of 5 mm. Significantly, higher speeds were reached when machining pure copper in the Li study [10], with twice as high Ra values.

A fully quadratic regression model was created. The insignificant factors (*p*-value > 0.05) were removed using the “Step Wise Selection of Terms” method while maintaining the hierarchy of the model. The correspondence between the cutting speed and thickness is an inverse proportion because the volume of material removed over time is more or less constant. Therefore, the model was calculated using the inverse thickness value.

The regression model for the cutting speed *v_c_* describes 99.47% of the variability of the monitored data. As shown in Figure 3, about 95% is described by the inverse value of the material thickness *t* alone. Therefore, the relationship for calculating the cutting speed is:(1)$$\begin{array}{l} v_{c}=4.126-0.429T_{on}-0.025T_{off}0.1485I+71.11\cdot 1/thickness-\\ -29.92\cdot 1/thickness\cdot 1/thickness+0.01487T_{on}\cdot I+3.657T_{on}\cdot 1/thickness+\\ +0.000961T_{off}\cdot I-1.6234T_{off}\cdot 1/thickness+1.857I\cdot 1/thickness. \end{array}$$

The significance of the regression model’s constituents for the cutting speed, including the contribution of each factor expressed as percentages, is shown in Table 3. Both the model as a whole (*p*-value_Regression < 0.05) and the individual terms are important (*p*-value < 0.05). The pulse off time was included to preserve the hierarchy in the model as its interactions are considerable.

In the graphs of the main effects (Figure 3a) and interactions (Figure 3b), the inverse of the material thickness is plotted as a factor, so the dependence is shown as linear, despite the relationship between the cutting speed and thickness being hyperbolic (inverse proportion). The inverse value of the thickness is the most important factor, as seen in the main effects graph. This quality was to be expected; however, for the inverse of the thickness to reach almost 95% variability of the model was very surprising. Other factors affect the cutting speed less (their impact altogether is only 4.5%). They are, however, statistically relevant as seen in the previous regression model in Table 3. If a specific thickness is chosen, its impact is fixed, which is why the setting is the only relevant model information. The pulse on time and discharge current have a positive effect while the pulse off time has a negative impact. The magnitude of the interaction in the graphs in Figure 3b can be understood from the nonparallel lines describing how the second observed factor influences the cutting speed. For practical use, the most interesting fact to notice is that the interaction with the inverse of the thickness only makes sense for small thicknesses. For greater thicknesses, the pulse on and off time, discharge current, and cutting speed are constant.

A fully quadratic regression model was created for the Ra_center. The insignificant factors (*p*-value > 0.05) were removed using the “Step Wise Selection of Terms” method while maintaining the hierarchy of the model. It does not make sense to use the inverse of the thickness value, so the thickness was left in a linear form, as is common for regression models.

The regression model for the Ra_ center describes 32.32% of the variability of the monitored data, which is common due to the large variability of Ra in WEDM surfaces. This model is clearly nonideal, but it contains significant parameters of the regression equation. The remaining variability is due to random effects, such as the material itself, and the factors, which were not investigated. The relationship for the Ra_center is as follows:(2)$$\begin{array}{l} Ra\_centre=-7.254+0.0367T_{on}+0.2329T_{off}+0.3115I+0.00632thickness-\\ -0.000029thickness\cdot thickness-0.007531T_{off}\cdot I. \end{array}$$

The significance of the regression model’s terms, including the contribution of each factor expressed as percentages, is shown in Table 4.

There is a statistically significant quadratic curvature of the Ra_center concerning thicknesses, as seen in the main effects graph shown in Figure 4a. All monitored factors have positive main effects, i.e., they cause an increase in the topography parameter. The pulse off time (µs)*discharge current (A) interactions have the most significant effect, as can be seen from the interaction graph in Figure 4b. This graph shows that when the discharge current is set to the lower level (25 A), an increase in the pulse off time causes an increase in the Ra_center. In contrast, when the discharge current is set to the upper level, an increase in the pulse off time causes a decrease in the Ra_center. The 3D sample’s relief of 5 and 160 mm thickness is shown in Figure 4c,d.

A screening experiment was run for the 10, 70, and 130 mm thicknesses preceding the creation of this model. The data from the original experiment was not used to calculate the models for the cutting speed and Ra_center, but it was used for its validation. All the observations of both responses lay within 95% of the prediction interval for the corresponding setting of the entry-level predictors. The model can be considered validated.

The edge roughness of the Ra_margin is always better (smaller) than that of the Ra_center, therefore, we do not present the regression model, and we only present the overall summary for the given response of the Ra_margin, which is shown in Figure 5.

The actual machining parameters for each thickness will be the output of a multi-criteria optimization for the cutting speed and Ra_center. The input to this optimization is the above regression models and response requirements. The optimization is implemented in MINITAB 19 using the response optimizer procedure. The article describes mathematical models for optimization, but the multicriteria optimization itself depends on the chosen response weight and is counted for specific fixed thicknesses. It is impossible to state the optimum as it is a function of the chosen weight for the cutting speed, Ra_center, and thickness. Similar optimization techniques were also used in the studies undertaken by Chaudhari et al. [24] and Chaudhari et al. [25].

### 3.2. Surface and Subsurface Area Analysis

A typical morphology for a WEDM-machined surface is formed by random craters. These occur due to the electrical discharges, which are necessary for material separation. Several studies have focused on the craters themselves, such as those by Vignesh et al. [26], Esteves et al. [27] and Zhang et al. [28]. The shape of the craters, their size, and frequency vary depending on the machined material, its heat treatment, and the machining parameters. These dependencies have been demonstrated in many previous studies, such as those by Mouralova et al. [29] and Altuğ [30]. Due to the very high temperatures of 10,000–20,000 °C [31] resulting from the eroding process, it is common for the upper layer of the machined material to be covered with a heat-affected recast layer. The recast layer melts completely and then rapidly cools as it comes into contact with the dielectric fluid. Certain materials and heat treatments are affected by surface and subsurface defects through this process, as described in several studies, such as those by Reddy et al. [32] and Chaudhari et al. [33].

In all cases, a secondary electron (SE) detector was used for imaging, and the samples were always studied at a magnification of 1000× and then 2500×. The surface morphology analysis of the samples did not reveal any of the possible defects, as shown in Figure 6 and Figure 7. There are only slight differences between the centers of the samples and the margins, particularly in the size and articulation of each crater, which completely corresponds to the topography analysis. Furthermore, as expected and proven by the topography analysis, both edges were found to have a similar appearance. The morphology of the samples mostly consists of smooth surfaces and crater bottoms. The surface is not especially rugged, just like in the case of Nitinol [34] machining. In contrast, a highly rugged surface was studied while machining stainless steel [35] and Ti6Al4V titanium alloy [36]. A smooth surface is ideal for the correct functionality and predicted tool life of the manufactured parts. Defects present in the subsurface layer would mean a significant reduction in the service life of the mold parts or their incorrect functioning from the very beginning of their operation in the mold.

Locations covered with segregated lead needles were found at the margins of several samples (especially those of greater thicknesses) (Figure 7a), as confirmed by the chemical composition analysis shown in Figure 8. Lead is a part of the base material, which is also evident from the image of the microstructure of the material shown in Figure 1d. It formed these needles due to the high temperatures during the cutting process. Similar phenomena were investigated in a study by Mouralova et al. [22] when machining the same material. These needles cause no issues for the mold part produced.

The sample’s cross-sectional analysis was performed on prefabricated metallographic preparations. These were examined by means of electron microscopy using a backscattered electron (BSE) detector with a magnification of initially 1000× and later 2500×. No defects were found on any sample or in a different margin-to-center position. The surface is covered in a layer of adhesive, as shown in Figure 9, but there are no cracks or burnt cavities. This is an important discovery because it means that defects are unlikely to affect the cutting speed optimization or surface topography. This defect-free state of the subsurface layer was also studied on the material. The formation of defects in the form of burnt cavities is caused by high temperatures at the cutting point. In the dielectric bath of the machine, the water dissociates, and atomic hydrogen is diffused under the surface of the machined material. Defects in the form of fissures are caused by high residual voltage in the upper layer of the workpiece material.

## 4. Conclusions

For this extensive Box–Behnken-type design experiment, a total of 448 samples with thicknesses ranging from 5 to 160 mm with a step of 5 mm were produced using WEDM. The studied material was Ampcoloy 35, which is used for the production of plastic injection mold parts. We analyzed, measured, and studied the topography of the machined surfaces in the center and at the margins of the samples, as well as the cutting speed, morphology, and subsurface layer. The following conclusions were reached:-The highest value of Ra 4 µm and a value of 3.6 were measured, respectively, in the center of the samples and at the margins. The lowest value of Ra 1.6 µm was measured in the sample machined with the following parameters: *T_on_* = 6 µs, *T_off_* = 40 µs, and *I* = 25 A;-All monitored factors have positive main effects, i.e., they cause an increase in the topography parameter Ra, with the pulse off time (µs)*discharge current (A) interaction as the most influential one;-True to our expectations, the highest speed of 20.24 mm/min was observed in the sample with the lowest material thickness of 5 mm;-The pulse on time and discharge current were found to have a positive effect, and the pulse off time to hurt the cutting speed;-The analysis of the surface morphology of the samples did not reveal any of the possible defects. There are only slight differences between the samples’ centers and margins in the form of the size and articulation of the craters, which fully corresponds with the topography analysis;-Spots covered with segregated lead needles were found at the margins of several samples (especially those of greater thickness);-The cross-sectional analysis of the samples showed no defects, even at different margin-to-center positions.

The conclusions show that the provided regression models for cutting speed and surface topography allow the efficient defect-free machining of Ampcoloy 35 of 5–160 mm thicknesses. The next goal is to test and optimize the thicknesses up to 250 mm. As of now, the 160 mm thickness represents the only restraint in the optimized production of mold parts and molds themselves, as it is necessary to machine thicknesses above 160 mm as well.

## Figures and Tables

**Figure 1 materials-16-00100-f001:**
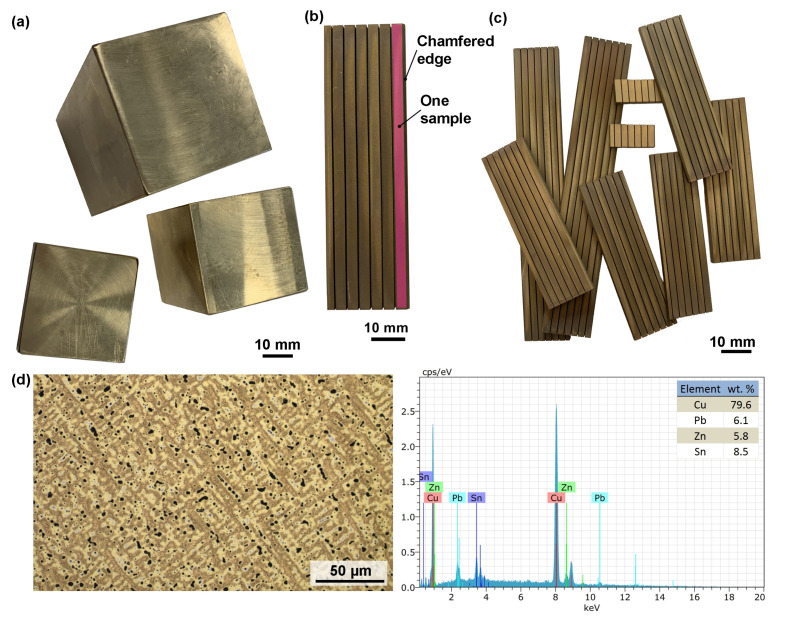
(**a**) Example of semifinished products for sample production, (**b**) example of a produced plate with 6 samples, (**c**) example of produced plates, (**d**) microstructure and EDX analysis.

**Figure 2 materials-16-00100-f002:**
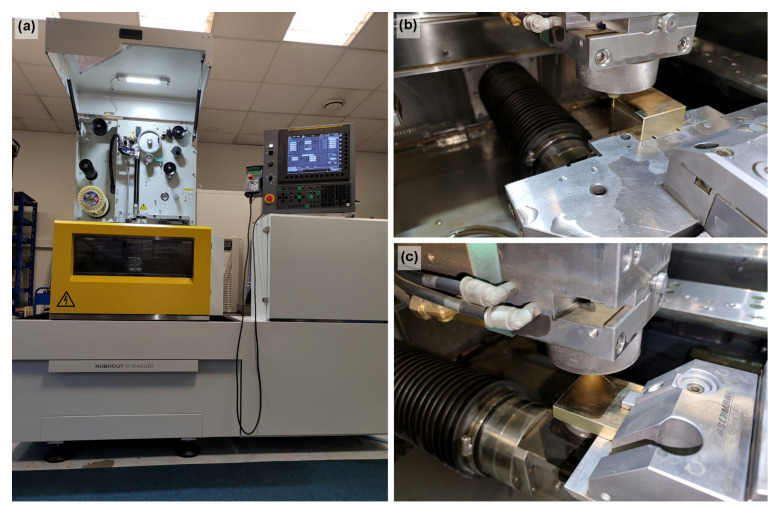
(**a**) Wire-cutting machine FANUC, (**b**,**c**) sample production.

**Figure 3 materials-16-00100-f003:**
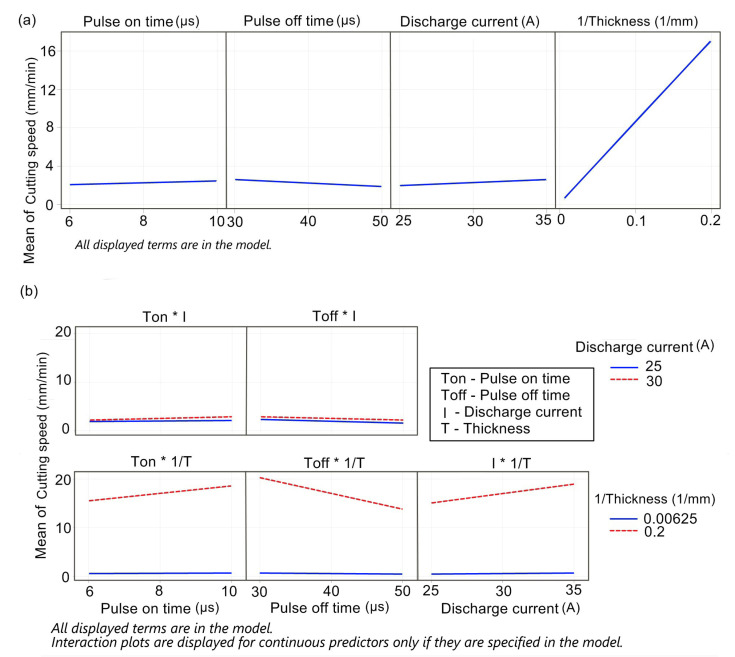
(**a**) Main effects of cutting speed, (**b**) interaction of cutting speed.

**Figure 4 materials-16-00100-f004:**
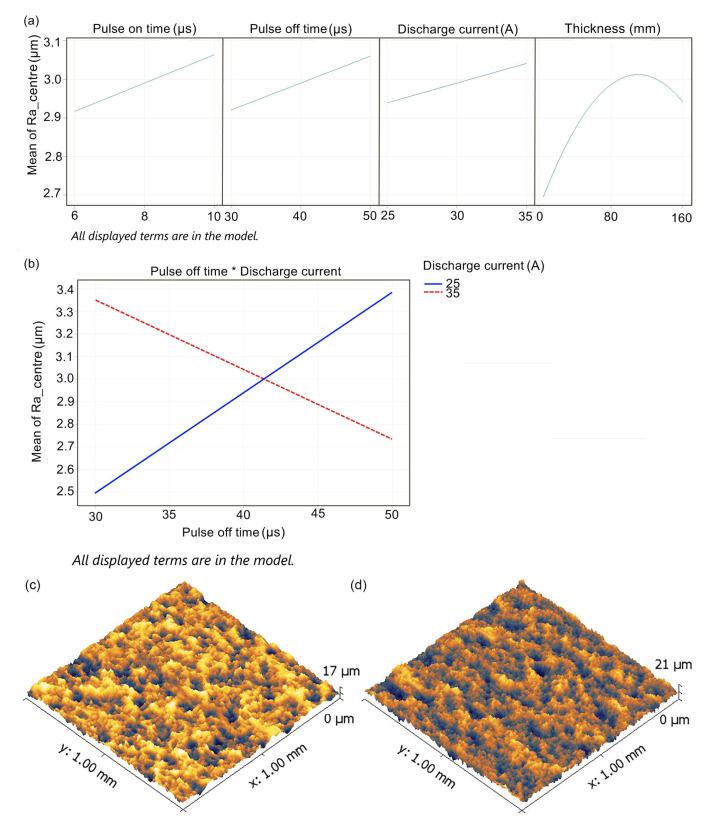
(**a**) Main effects of Ra_center, (**b**) graph interactions for Ra_center, (**c**) 3D relief of the sample’s surface at 5 mm thickness, (**d**) 3D relief of the sample’s surface at 160 mm thickness.

**Figure 5 materials-16-00100-f005:**
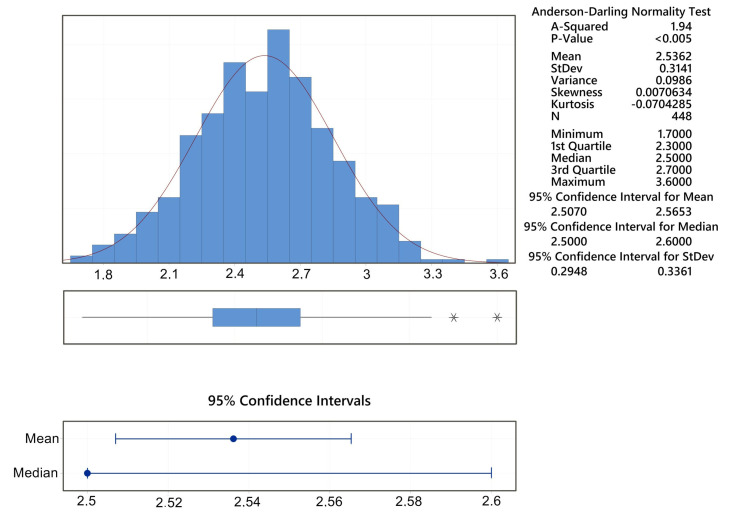
Summary graph for Ra_margin.

**Figure 6 materials-16-00100-f006:**
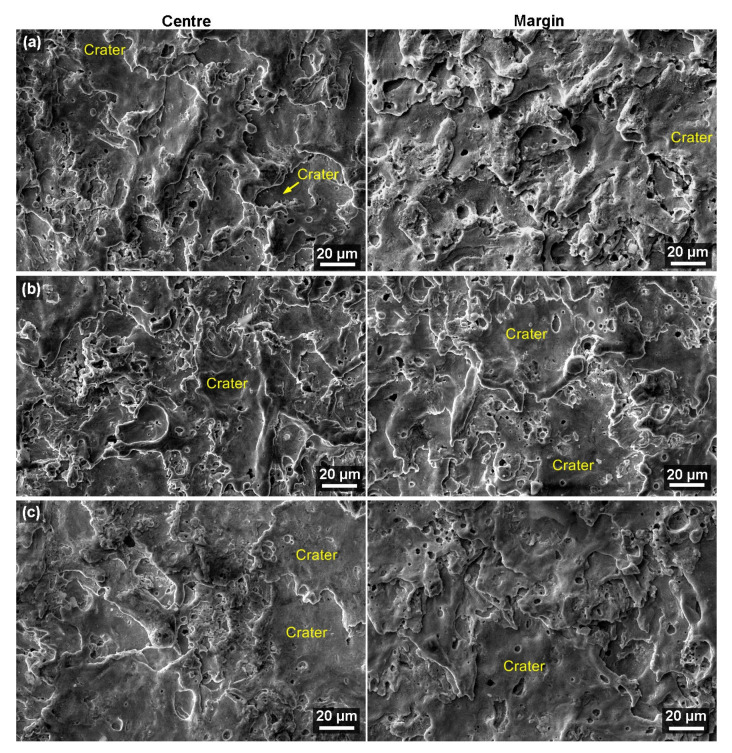
Surface morphology of individual samples at the margin or in the center of SEM/SE, (**a**) sample with a thickness of 5 mm, (**b**) sample with a thickness of 35 mm, (**c**) sample with a thickness of 70 mm.

**Figure 7 materials-16-00100-f007:**
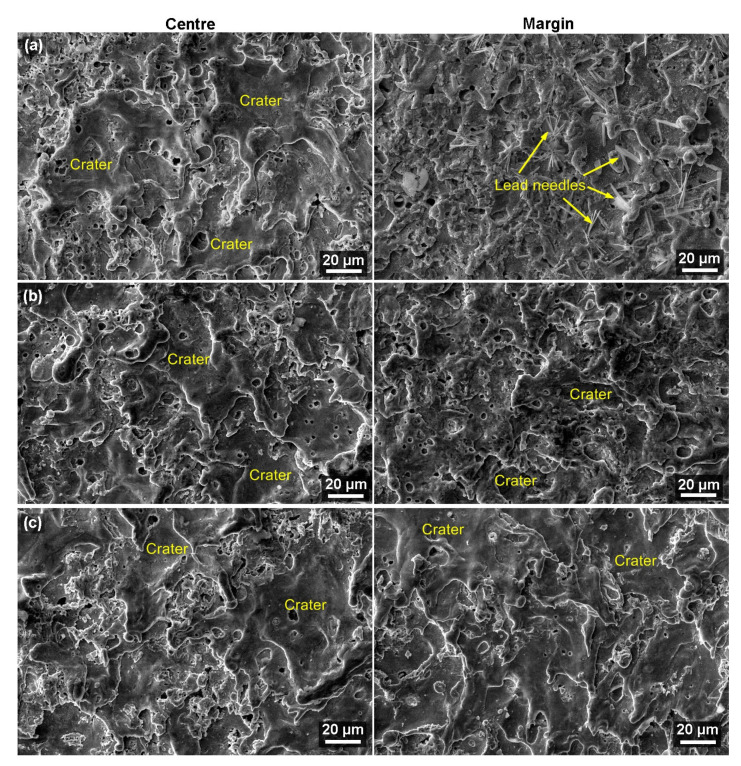
Surface morphology of samples at the margins and in the center of SEM/SE, (**a**) sample with a thickness of 100 mm, (**b**) sample with a thickness of 135 mm, (**c**) sample with a thickness of 160 mm.

**Figure 8 materials-16-00100-f008:**
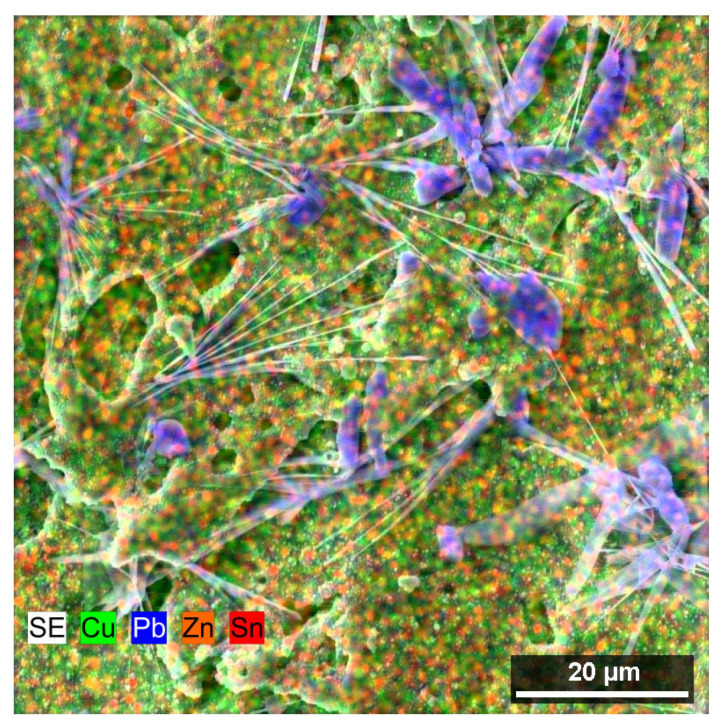
Distribution map of the individual elements in the area of occurrence of SEM/SE needles.

**Figure 9 materials-16-00100-f009:**
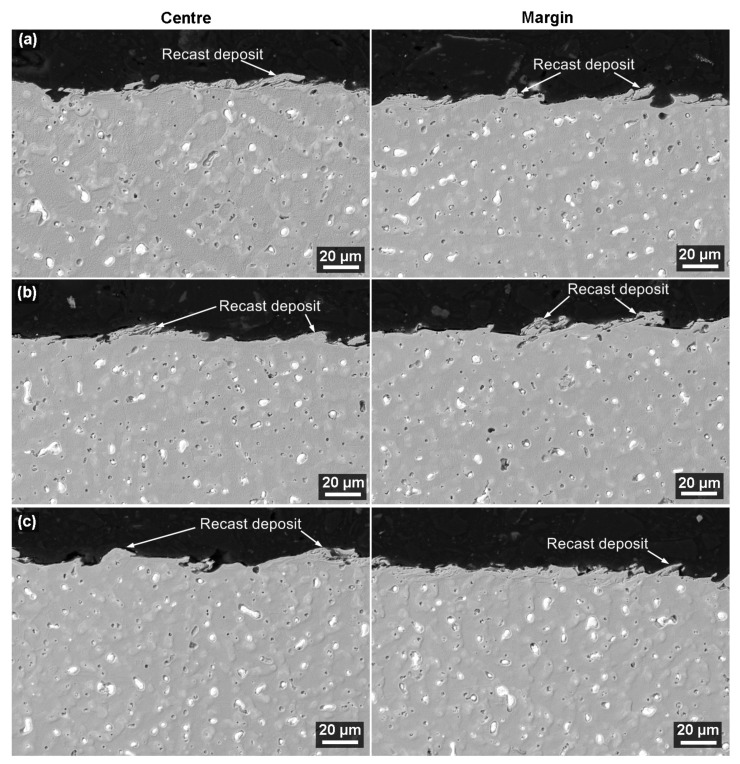
Cross-section of the sample in the center and margin positions SEM/BSE, (**a**) sample with a thickness of 5 mm, (**b**) sample with a thickness of 80 mm, (**c**) sample with a thickness of 160 mm.

**Table 1 materials-16-00100-t001:** The design of the experiment for one material thickness.

Std Order	Run Order	Pt Type	Pulse on Time (µs)	Pulse off Time (µs)	Discharge Current (A)
3	1	2	6	50	30
9	2	2	8	30	25
7	3	2	6	40	35
8	4	2	10	40	35
6	5	2	10	40	25
12	6	2	8	50	35
11	7	2	8	30	35
13	8	0	8	40	30
14	9	0	8	40	30
5	10	2	6	40	25
1	11	2	6	30	30
4	12	2	10	50	30
2	13	2	10	30	30
10	14	2	8	50	25

**Table 2 materials-16-00100-t002:** Ranges of setup parameters.

Level	Pulse on Time (µs)	Pulse Off Time (µs)	Discharge Current (A)
High	10	50	25
Middle	8	40	30
Low	6	30	35

**Table 3 materials-16-00100-t003:** Regression model of cutting speed, including the expressed percentage contribution of individual factors.

Analysis of Variance			
Source	Contribution	F-Value	*p*-Value
Regression	99.47%	8239.57	0.000
Pulse on time (µs)	0.26%	44.59	0.000
Pulse off time (µs)	0.75%	3.78	0.053
Discharge current (A)	0.42%	37.51	0.000
1/Thickness (1/mm)	94.96%	418.57	0.000
1/Thickness (1/mm)*1/Thickness (1/mm)	0.04%	33.82	0.000
Pulse on time (µs)*Discharge current (A)	0.06%	49.19	0.000
Pulse on time (µs)*1/Thickness (1/mm)	0.40%	327.22	0.000
Pulse off time (µs)*Discharge current (A)	0.01%	5.14	0.024
Pulse off time (µs)*1/Thickness (1/mm)	1.95%	1612.19	0.000
Discharge current (A)*1/Thickness (1/mm)	0.64%	527.53	0.000
Error	0.53%		
Total	100.00%		

**Table 4 materials-16-00100-t004:** Regression model of Ra_center.

Analysis of Variance
Source	Contribution	F-Value	*p*-Value
Regression	32.32%	35.09	0.000
Pulse on time (µs)	1.77%	11.53	0.001
Pulse off time (µs)	1.60%	158.91	0.000
Discharge current (A)	0.86%	157.16	0.000
Thickness (mm)	3.11%	18.72	0.000
Thickness (mm)*Thickness (mm)	1.71%	11.13	0.001
Pulse off time (µs)*Discharge current (A)	23.27%	151.59	0.000
Error	67.68%		
Total	100.00%

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
