# Peer review of "Mathematical Models for Machining Optimization of Ampcoloy 35 with Different Thicknesses Using WEDM to Improve the Surface Properties of Mold Parts"

_materials, 2022, doi:10.3390/ma16010100_

Round 1

Reviewer 1 Report

The article deals with optimization of the WEDM process for the machining of Ampcoloy 35 in terms of maximizing cutting speed and surface quality. WEDM is often necessary in the production of molds where classical machining is impossible. The authors established regression models for cutting speed and surface roughness. They carried out the sufrace morphology of individual samples. The cross-sectional analysis of the samples showed no defects what also confirms the correctness of the experiment procedure. I highlight the number of samples the authors worked with. Conclusions are consistent. The article contains a small number of self-citations, but they refer to the authors' related previous research.

Provided regression model allows effcient machining of Ampcoloy 35.

I have few comments:

1. I think that chapter 3.1 should be included in previous chapter "Experimental setup and materials"

2. In chapter 3.2 on page 7, authors write that the values measured at the margin and in the centre of the samples were recorded in Table 2. Is this correct? Tab. 2 includes "Rangers of setup parameters", I think it has to be "Ranges" not "Rangers"

3. Chapter 3.3 begins next to figure 5.

Author Response

The article deals with optimization of the WEDM process for the machining of Ampcoloy 35 in terms of maximizing cutting speed and surface quality. WEDM is often necessary in the production of molds where classical machining is impossible. The authors established regression models for cutting speed and surface roughness. They carried out the sufrace morphology of individual samples. The cross-sectional analysis of the samples showed no defects what also confirms the correctness of the experiment procedure. I highlight the number of samples the authors worked with. Conclusions are consistent. The article contains a small number of self-citations, but they refer to the authors' related previous research.

Provided regression model allows effcient machining of Ampcoloy 35.

I have few comments:

  1. I think that chapter 3.1 should be included in previous chapter "Experimental setup and materials"

Thank you, I moved this chapter 3.1.

  1. In chapter 3.2 on page 7, authors write that the values measured at the margin and in the centre of the samples were recorded in Table 2. Is this correct? Tab. 2 includes "Rangers of setup parameters", I think it has to be "Ranges" not "Rangers"

You are right, it was my mistake and I delete whole sentence. Table with measured values is very long and this journal is not allow it. I also corrected word Ranges.

  1. Chapter 3.3 begins next to figure 5.

It was caused only by transforming to pdf version. In doc version of manuscript it is correct, thank you.

Thank you for your time and all comments!

Reviewer 2 Report

the author do a lots of experiments, however the mathematic models is just a regression model. Hence this paper is lack of deep analysis and innovation in my opinion. The author should pay attention to discuss the experimental results, and given deep analysis.

Author Response

the author do a lots of experiments, however the mathematic models is just a regression model. Hence this paper is lack of deep analysis and innovation in my opinion. The author should pay attention to discuss the experimental results, and given deep analysis.

Thank you for your comment. There is not optimization because this experiment continued also for thicknesses 165-250 mm and in the end of it is also optimization part. This manuscript focused on regresion model and also on Surface and subsurface area analysis that is for WEDM very important.

I discussed my results with other results in many cases, for example: "These values are higher than the maximum Ra values observed in a previous study done by Mouralova et al. [19] when machining the same material of a 10 mm thickness.", " . Similarly low values were obtained for copper machining in the studies done by Venkateswarlu et al. [20] and Satishkumar et al.[8]. ", "Significantly higher speeds were reached when machining pure copper in the Li study [9], with twice as high Ra values.", "Similar optimization techniques were also used in studies done by Chaudhari et al.[32] or Chaudhari et al. [33].", "The surface is not especially rugged, just like in the case of Nitinol [29] machining. In contrast, a highly rugged surface was studied while machining stainless steel [30] and Ti6Al4V titanium alloy [31]."...

Thank you for your time and all comments!

Reviewer 3 Report

The manuscript makes an in-depth and detailed study of the technological research of WEDM. The experimental method is correct, the experimental data are sufficient, and the conclusions are correct, which is in line with the positioning of the journal. It is recommended to accept the publication.

1. There is no consistency in the form of English punctuation. In particular, commas.

2. In many places there is no space after the punctuation mark.

Author Response

The manuscript makes an in-depth and detailed study of the technological research of WEDM. The experimental method is correct, the experimental data are sufficient, and the conclusions are correct, which is in line with the positioning of the journal. It is recommended to accept the publication.

  1. There is no consistency in the form of English punctuation. In particular, commas.

Thank you for your comment, foreign speaker revised whole manuscript.

  1. In many places there is no space after the punctuation mark.

Thank you, I hope that I fixed all these places.

Thank you for your time and all comments!

Reviewer 4 Report

Thanks for submitting the manuscript. Following are my observations:

Thermal conductivity which is 45 W·m−1·K−1, and a relatively good hardness up to 450 HB.

Ampco alloys include high thermal conductivity up to 208 W·m− 1·K− 1

Check which is correct with reference.

Model of EDM optimization

Process WEDM

Material Ampcoloy35

Process parameters: Pulse on and off time, Discharge current, and material thickness

Output: Cutting speed, topography of the machined surfaces in the centre and at the margins of the sample, morphology and subsurface layer

Extensive experiments have been performed and regression model was set on the process parameters for the process response. In the title is was stated that a mathematical model was being developed for the optimization. An analytical model as expected from the title but a regression model, which does not contain any physics of the process was produced. Therefore, there is no novelty in the work. The level of experiments is not more than a post graduate project work. The present work is not adding any knowledge to the literature. If an analytical model can be developed on this process, then this can be archived as research document. In addition, thermal models are available starting from the heat flux to the material removal. Advancement in the model may be reported to be a good research work. Kindly understand, repeating the same type of work by changing the material only may not be considered as a research work.

Concluding remark:

Make the job crisp, short, interesting and something novel. It looks reading same type of job again and again.

Few points are given, that may be useful for improving the quality of the document:

1.       What is the value addition of this statement in the abstract “Copper alloys are commonly used for their high thermal conductivity”, is it for electrode or supplying electricity to the instrument?

2.       5+5 up to 160 mm thickness total 448 samples have been prepared.

3.       Reference of material properties, such as thermal conductivity and hardness.

4.       Duty cycle 12% and current is very high up to 30 amp. The on off time and the current value must be crosschecked. It is found from the experience that many times the EDM suppliers don’t provide the correct value.

Author Response

  1. What is the value addition of this statement in the abstract “Copper alloys are commonly used for their high thermal conductivity”, is it for electrode or supplying electricity to the instrument?

Thank you for your comment I added to abstract: "Copper alloys are commonly used as a electrodes for their high thermal conductivity".

  1. 5+5 up to 160 mm thickness total 448 samples have been prepared.

Yes, it is true. This research was supported by Technology Agency of the Czech Republic, project no. FW03010044. And title of this project is: "Improvement of surface properties of mold parts produced by WEDM technology in order to increase production productivity and reduce or eliminate the use of chemical mold separators." There are two outcomes - one is Verified technology and second is Funkcional sample.

  1. Reference of material properties, such as thermal conductivity and hardness.

Thank you, I added reference [36].

  1. Duty cycle 12% and current is very high up to 30 amp. The on off time and the current value must be crosschecked. It is found from the experience that many times the EDM suppliers don’t provide the correct value.

Thank you, I know it, thats why was also used oscilloscope for checking. All was correct.

Thank you for your time and all comments!

Reviewer 5 Report

Yes, I agree with the authors of the manuscript that WEDM optimization is a complex task because there are a large number of machining parameters, including the thickness of the material being machined, that need to be optimized separately. Optimizing the process brings significant savings in machining time and thus also a reduction in the amount of electrical energy consumed for machining a specific part. At the same time, it allows to increase the efficiency of the process. Therefore, from this point of view, I consider the proposal of experimental research to be very relevant. Although it is questionable whether the process optimization for machining Ampcoloy 35 with WEDM technology was sufficiently described in the work. Even so, I consider the information obtained from the results of experimental research to be valuable.

Although the manuscript is well written, some corrections need to be made and the following questions answered:

1. It is generally known that the roughness of the machined surface after WEDM depends on the combination of setting the main technological parameters, of which Ton, Toff, I, U and others have a significant position. When evaluating the influence of a specific input technological parameter in the WEDM process on the roughness of the machined surface, it is necessary that the values ​​of the other parameters are constant during the machining process. Were these conditions met during the experiment?

2. In the abstract it is written "The subject of this study was the optimization of WEDM machining of the copper alloy Ampcoloy 35 of a thickness ranging from 5 to 160 mm with a step of 5 mm." However, the optimization is missing in the manuscript. So either correct the abstract and the title of the manuscript, or complete the optimization. At the same time, the abstract is written very generally. It is necessary to write it more specifically with the justification of why the given research was carried out. What is the research gap in the given area and what results did the authors achieve through the experimental research they carried out.

3. How was the surface roughness parameters Ra measured in the narrow cutting slot?

4. Are the stated results of experimental findings applicable to the machining of other materials using the given cutting tools and machining conditions? This should be discussed at the end of the manuscript.

5. At the end of the thesis, there is no discussion in which the authors evaluate the conclusions in the context of existing knowledge. At the conclusion of the thesis, it is very important to qualitatively or quantitatively emphasize the points of agreement or disagreement between the results in this thesis and the cited references in the manuscript.

The contribution is processed at a good level and after modifications and additions it can be published in the journal Materials.

Author Response

Although the manuscript is well written, some corrections need to be made and the following questions answered:

  1. It is generally known that the roughness of the machined surface after WEDM depends on the combination of setting the main technological parameters, of which Ton, Toff, I, U and others have a significant position. When evaluating the influence of a specific input technological parameter in the WEDM process on the roughness of the machined surface, it is necessary that the values ​​of the other parameters are constant during the machining process. Were these conditions met during the experiment?

Thank you for your comment, I know that this is very important factor. I always take care of it and I rather check it twice. It was always set up the same even though number of samples is huge.

  1. In the abstract it is written "The subject of this study was the optimization of WEDM machining of the copper alloy Ampcoloy 35 of a thickness ranging from 5 to 160 mm with a step of 5 mm." However, the optimization is missing in the manuscript. So either correct the abstract and the title of the manuscript, or complete the optimization. At the same time, the abstract is written very generally. It is necessary to write it more specifically with the justification of why the given research was carried out. What is the research gap in the given area and what results did the authors achieve through the experimental research they carried out.

Thank you, I improved manuscript and now is:” Wire electrical discharge machining (WEDM) is an unconventional machining technology that can be used to machine materials with a minimum electrical conductivity. The technology is often employed in the automotive industry, as it makes it possible to produce mould parts of complex shapes. Copper alloys are commonly used as a electrodes for their high thermal conductivity. The subject of this study was creating mathematical models for machining optimization  of Ampcoloy 35 with  different thicknesses (ranging from 5 to 160 mm with a step of 5 mm) using WEDM to improve the surface properties of  mould parts. The Box-Behnken type experiment was used with a total of 448 samples produced. The following machining parameters were altered over the course of the experiment: Pulse on and off time, Discharge current, and material thickness. The cutting speed was measured and the topography of the machined surfaces in the centre and at the margins of the samples, were analysed. The morphology and subsurface layer were also studied. What makes this study unique is the large number of the tested thicknesses, ranging from 5 to 160 mm with a step of 5 mm. The contribution of this study to the automotive industry and plastic injection mould production is therefore significant. The regression models for cutting speed and surface topography allow for efficient defect-free machining of Ampcoloy 35 of 5-160 mm thicknesses, both in the surface and subsurface layer.

  1. How was the surface roughness parameters Ra measured in the narrow cutting slot?

There were not measured. Slot is there for separation individual samples.

  1. Are the stated results of experimental findings applicable to the machining of other materials using the given cutting tools and machining conditions? This should be discussed at the end of the manuscript.

No, there are not. That's why is always necessary to do experiments for every material and also its heat treatment if any.

  1. At the end of the thesis, there is no discussion in which the authors evaluate the conclusions in the context of existing knowledge. At the conclusion of the thesis, it is very important to qualitatively or quantitatively emphasize the points of agreement or disagreement between the results in this thesis and the cited references in the manuscript.

For this reason is there chapter 3. Results and discussion where I discussed my results with results in other studies.For example: "These values are higher than the maximum Ra values observed in a previous study done by Mouralova et al. [19] when machining the same material of a 10 mm thickness.", " . Similarly low values were obtained for copper machining in the studies done by Venkateswarlu et al. [20] and Satishkumar et al.[8]. ", "Significantly higher speeds were reached when machining pure copper in the Li study [9], with twice as high Ra values.", "Similar optimization techniques were also used in studies done by Chaudhari et al.[32] or Chaudhari et al. [33].", "The surface is not especially rugged, just like in the case of Nitinol [29] machining. In contrast, a highly rugged surface was studied while machining stainless steel [30] and Ti6Al4V titanium alloy [31]."... Chapter "Conclusion" summarize results in clear way for all readers.

The contribution is processed at a good level and after modifications and additions it can be published in the journal Materials.

Thank you for your time and all comments!

Round 2

Reviewer 4 Report

thanks for the corrections as suggested.